# Diversity and Differential Expression of MicroRNAs in the Human Skeletal Muscle with Distinct Fiber Type Composition

**DOI:** 10.3390/life13030659

**Published:** 2023-02-28

**Authors:** Andrey V. Zhelankin, Liliia N. Iulmetova, Ildus I. Ahmetov, Eduard V. Generozov, Elena I. Sharova

**Affiliations:** 1Department of Molecular Biology and Genetics, Lopukhin Federal Research and Clinical Center of Physical-Chemical Medicine of Federal Medical Biological Agency, 119435 Moscow, Russia; 2Research Institute for Sport and Exercise Sciences, Liverpool John Moores University, Liverpool L3 5AF, UK

**Keywords:** microRNA, miRNome, transcriptome, muscle fiber type, power athlete, endurance athlete

## Abstract

The ratio of fast- and slow-twitch fibers in human skeletal muscle is variable and largely determined by genetic factors. In this study, we investigated the contribution of microRNA (miRNA) in skeletal muscle fiber type composition. The study involved biopsy samples of the *vastus lateralis* muscle from 24 male participants with distinct fiber type ratios. The miRNA study included samples from five endurance athletes and five power athletes with the predominance of slow-twitch (61.6–72.8%) and fast-twitch (69.3–80.7%) fibers, respectively. Total and small RNA were extracted from tissue samples. Total RNA sequencing (*N* = 24) revealed 352 differentially expressed genes between the groups with the predominance of fast- and slow-twitch muscle fibers. Small RNA sequencing showed upregulation of miR-206, miR-501-3p and miR-185-5p, and downregulation of miR-499a-5p and miR-208-5p in the group of power athletes with fast-twitch fiber predominance. Two miRtronic miRNAs, miR-208b-3p and miR-499a-5p, had strong correlations in expression with their host genes (*MYH7* and *MYH7B*, respectively). Correlations between the expression of miRNAs and their experimentally validated messenger RNA (mRNA) targets were calculated, and 11 miRNA–mRNA interactions with strong negative correlations were identified. Two of them belonged to miR-208b-3p and miR-499a-5p, indicating their regulatory links with the expression of *CDKN1A* and *FOXO4*, respectively.

## 1. Introduction

Skeletal muscle is a highly plastic tissue composed of fiber types, which differ in structure, molecular composition and functional properties. Adult skeletal muscle fibers are broadly classified as slow-twitch (type I) and fast-twitch (types IIa and IIx). Type I and IIa fibers primarily use oxidative metabolism, while type IIx are more reliant on glycolytic metabolism [1,2]. The basis of molecular and functional heterogeneity of human muscle fibers can be identified in the diversity of protein composition and gene expression patterns [3]. Distinct mechanical and energetic properties of skeletal muscle fibers are due to differences in the molecular composition of myofibrils, the intracellular structures containing the molecular motor of muscle contraction. Fast and slow muscle fibers have differences in the expression of myofibrillar protein isoforms, myosin binding proteins, thin filament proteins, nebulin and actin-binding proteins [4,5,6,7,8]. Differences are also found in the expression of proteins associated with excitability and excitation–contraction coupling, electrical membrane properties and ion channels, calcium release and energy production systems [7,9]. The main techniques for muscle fiber typing are histochemical staining for myosin ATPase, myosin heavy chain isoform identification and biochemical identification of metabolic enzymes [10].

The heterogeneity of muscle fiber composition is likely to have an impact on individual response to exercise training. The ratio of fast- and slow-twitch fibers in human skeletal muscle tissue is variable and largely determined by genetic factors [11,12]. There are many inherited myopathies and other acquired muscle-related disorders, which preferentially affect specific skeletal muscle fiber types [13]. At the same time, some studies indicate that the type of physical activity as well as its reduction can slightly shift the fiber type composition [13,14,15,16]. The predominance of endurance training induces the transition from fast-twitch to slow-twitch muscle fiber, whereas power training has the opposite effect [2,14,17]. Various components of calcium-dependent signaling pathways and multiple transcription factors, coactivators and corepressors have been shown to be involved in skeletal muscle remodeling [18]. Fiber type shifting is regulated by myogenic regulatory factors (MYF5, MYOD, myogenin and MRF4), myocyte enhancer factor-2 (MEF2), calcineurin and nuclear factor of activated T cells (NFAT), peroxisome proliferator-activated receptor (PPAR)-γ coactivator-1α (PGC-1α), Ca^2+^-dependent transcription factor and transient receptor potential melastatin (TRPM2)-mediated Ca^2+^ signaling [2,17,19,20,21,22,23].

Non-coding RNAs are well known to participate in a variety of important regulatory processes in myogenesis. One of the tools for regulating gene expression is post-transcriptional regulation by microRNAs (miRNAs)—a class of highly conserved single-stranded non-coding RNAs with a length of 21–24 nucleotides. Their function is realized by directly degrading messenger RNA (mRNA) of target genes or inhibiting target gene translation. Thus, miRNAs orchestrate the regulation of their targets to control the signaling pathways and common biological functions. In recent years, a number of studies have highlighted the importance of miRNAs in the control of skeletal muscle development and function through their influence on multiple biological signaling pathways, which are important for skeletal muscle homeostasis [24,25]. Alteration of the expression of many miRNAs or genetic mutations of miRNA genes are associated with changes in myogenesis and the progression of several skeletal muscle diseases [26]. A group of miRNAs known as myomiRs includes muscle-specific or muscle-enriched miRNA species, namely miR-1, miR-133a/b, miR-206, miR-208a/b, miR-486 and miR-499, which play a crucial role in myogenesis and muscle function [27,28,29,30]. This role is implemented in various forms of interaction between miRNAs and gene expression. Generally, some miRNAs are seen as playing key roles during myogenesis, e.g., miR-1/miR-206 or miR-133, while others likely constitute a kind of muscle property “fine-tuner”, including, e.g., miR-208a/b, which influences muscle performance by myosin switching [28,31]. The expression of some myomiRs is controlled by a set of muscle-specific transcription factors and cofactors, referred to as myogenic regulatory factors (MRFs) [29,32]. Some myomiRs are intragenic, and their expression rate primarily depends on that of their host gene. Three myosin genes, *MYH6*, *MYH7* and *MYH7B*, encode related miRNAs (miR-208a, miR-208b and miR-499, respectively) within their introns, which, in turn, control muscle myosin content, myofiber identity and muscle performance [33]. MyomiRs are expressed in both cardiac and skeletal muscle, with the exception of miR-206, which is skeletal-muscle-specific, and miR-208a, which is cardiac-muscle-specific.

Fiber types are a conserved feature of vertebrate muscle, and differential expression of miRNAs has been previously found between fast and slow fibers in various vertebrate species. In fish, the expression of miR-499 was increased in slow fibers of pacu (*Piaractus mesopotamicus*) [34]. In both fast and slow fibers, the expression of miR-1, miR-133a/b and miR-206 increased during fish growth. At the same time, the expression of their potential target genes involved in myogenesis (*hdac4*, *srf* and *pax7*, respectively) decreased with growth. Analysis of miRNA expression in bovine skeletal muscle by massively parallel sequencing found increased expression of miR-1, miR-133a/b and miR-206, and decreased expression of miR-208a in fast-type muscles compared to slow-type ones [35]. In recent studies, RNA sequencing has been applied to analyze transcriptomic differences and miRNA–mRNA interactions between muscles with a prevalence of fast and slow fibers in chickens, horses and donkeys [36,37,38]. In horses, the expression of miR-499 and miR-206 was elevated in muscles with a predominance of slow fibers. The authors suggest a co-regulatory model of regulating the proportion of fast and slow fiber types based on the interactions of *Sox6* with *Myh7b* and myomiRs [37]. In donkeys, miR-208a and miR-499-3p were among the upregulated miRNAs in muscles with a predominance of slow fibers [38]. The common factor in all of the above vertebrate skeletal muscle miRNA studies is that the expression of either miR-499-3p/5p or miR-208a/b was elevated in muscles with a predominance of slow fibers. This is in agreement with previous research on mice showing a potential role of these myomiRs in the specification of muscle fiber identity by activating slow and repressing fast myofiber gene programs [33]. However, there have been no studies so far on the differences in miRNA profiles in human skeletal muscle with different fiber type composition.

This study first uses integrated miRNome and transcriptome analysis to compare human skeletal muscle tissue samples with different ratios of fast- and slow-twitch fibers. This is a preliminary exploratory study focused on the possible miRNA regulation of muscle transcriptional activity as part of a global multi-omics research project on the regulation of gene expression in human skeletal muscle. We investigated human *m. vastus lateralis* biopsy samples of athletes with a preference for power or endurance exercise. By obtaining complete miRNA profiles, we showed miRNA diversity in skeletal muscle and found differences in miRNA expression between samples with a predominance of fast- and slow-twitch fibers. We first described the miRNA isoform composition in human skeletal muscle samples. Connecting transcriptomic data to the analysis, we analyzed the expression of key genes specific to different fiber types, showed the association between miRNAs and gene expression in skeletal muscle and suggested the extent to which miRNAs may be involved in regulating the fiber type composition.

## 2. Materials and Methods

### 2.1. Participants and Ethical Approval

For this study, physically active male participants of Russian origin with the predominance of slow-twitch or fast-twitch fibers in the *vastus lateralis* skeletal muscle of more than 60% were selected from the previously reported muscle biopsy study (*N* = 151) [39]. The study was approved by the local ethics committee, and written informed consent was obtained from each study participant.

### 2.2. Muscle Biopsy Preparation

Samples of the *vastus lateralis* muscle were obtained with the Bergström needle biopsy procedure with aspiration under local anesthesia with 2% lidocaine solution. Biopsy samples were placed in screw-cap freezing tubes, immediately snap-frozen in liquid nitrogen and stored at −80 °C. The samples were further divided without thawing into two parts for muscle fiber typing and for RNA isolation.

### 2.3. Evaluation of Muscle Fiber Composition

Serial cross-sections (7 μm) were obtained from frozen samples using a microtome (Leica Microsystems, Wetzlar, Germany). The sections were thaw-mounted on polylysine glass slides, and myosin heavy chain (MHC) isoforms were identified by immunohistochemical analysis, as previously described [40]. Fibers stained in serial sections with antibodies against slow and fast isoforms were considered as hybrid fibers. The fiber cross-sectional area was evaluated using ImageJ software (version 1.38, NIH, Bethesda, Montgomery County, MD, USA). The percentage of fast and slow muscle fibers was calculated as the ratio of the number of stained fibers to the total fiber number. Examples of images of muscle sections for samples with a predominance of fast- and slow-twitch fibers are shown in Figure A1, Appendix B.

RNeasy Mini Fibrous Tissue Kit (Qiagen, Hilden, Germany) was used to isolate RNA from muscle tissue samples. Frozen tissue samples were placed in a box submerged in liquid nitrogen. Each sample was transferred without thawing on a sterile Petri dish placed on a frozen plastic ice pack. A piece of tissue with a weight of 10 mg was separated with a sterile scalpel and immediately placed in a 2 mL safe-lock microcentrifuge tube containing 300 µL of lysis buffer and one sterile stainless-steel bead with a diameter of 4 mm. Samples were homogenized using the TissueLyser II system (Qiagen, Hilden, Germany) by shaking twice for 2 min at 25 Hz. RNA samples were isolated according to the manufacturer’s guidelines. RNA concentration was measured using the Qubit spectrophotometer (Thermo Fisher Scientific, Waltham, MA, USA). RNA quality was assessed using the BioAnalyzer electrophoresis system and the BioAnalyzer RNA Nano assay (Agilent Technologies, Santa Clara, CA, USA). RNA integrity number (RIN) was calculated for each RNA sample. Only RNA samples with RIN > 7 were included in the study. Samples were stored at −80 °C until sequencing libraries were prepared.

### 2.4. Small RNA Isolation

MiRNeasy Mini Kit (Qiagen, Hilden, Germany) was used to isolate small RNA from muscle tissue samples. Tissue samples were prepared in the same way as for total RNA isolation. Each sample was immediately placed in a 2 mL safe-lock microcentrifuge tube containing 700 µL of QIAzol buffer and one sterile stainless-steel bead with a diameter of 4 mm. Samples were homogenized using the TissueLyser II system (Qiagen, Hilden, Germany) by shaking twice for 2 min at 25 Hz. RNA samples were isolated according to the miRNeasy Mini Kit guidelines. RNA concentration was measured using the Qubit spectrophotometer (Thermo Fisher Scientific, Waltham, MA, USA). The presence of small RNA fraction was assessed using the BioAnalyzer electrophoresis system and the BioAnalyzer Small RNA assay (Agilent Technologies, Santa Clara, CA, USA). Samples were stored at −80 °C until sequencing libraries were prepared.

### 2.5. Total RNA Sequencing

Total RNA samples were treated with the DNAse I using Turbo DNA-free Kit (Thermo Fisher Scientific, Waltham, MA, USA) according to the kit guidelines. Libraries for RNA sequencing (RNA-seq) were prepared using the NEBNext Ultra II Directional RNA Library Prep Kit for Illumina with the NEBNext rRNA Depletion Module (New England Biolabs, Ipswich, MA, USA). RNA libraries were sequenced on the HiSeq system (Illumina, San Diego, CA, USA) in a paired-end mode with the read length of 125.

### 2.6. Small RNA Sequencing

Libraries for small RNA sequencing were prepared using the NEBNext Small RNA Library Prep Set for Illumina (New England Biolabs, Ipswich, MA, USA). The amount of RNA for library preparation varied from ~300 to ~450 ng. Size selection of libraries was performed using the AMPure XP magnetic beads (Beckman Coulter Life Sciences, Brea, CA, USA). The libraries’ size and quantity were assessed using the BioAnalyzer electrophoresis system and the BioAnalyzer DNA HS assay (Agilent Technologies, Santa Clara, CA, USA). Libraries were pooled in an equimolar ratio and sequenced using the MiSeq system (Illumina, San Diego, CA, USA) in a single-end mode with the read length of 50.

### 2.7. Data Analysis

For the whole transcriptome analysis, quality control by FastQC [41] and MultiQC [42] was performed before and after adapter trimming by Cutadapt (version 3.3) [43] and quality filtering by trimmomatic (version 0.39) [44] for the whole dataset. Pseudoalignment was performed by salmon [45] for GRCh38 reference and Gencode.v37 transcriptome without alt haplotypes. Summarizing of the gene was performed by tximport [46] ignoring the transcripts’ version. The expression and dispersion of genes and transcripts were estimated by edgeR (version 3.28.1) [47] with offset scaling following the tximport approach. Differential expression for the whole transcriptome was realized by the quasi-likelihood F-test with FDR-adjusted *p*-values < 0.05 and |FC| > 1.5. For further analysis, CPM data from edgeR were used through filtering by maximal mean in any group > 4 CPM. Principal component analysis (PCA) was realized by PCAtools (version 2.6.0) on log2 transformed CPM by removing genes with the lowest 10% variance [48]. Volcano plots were plotted with ggplot2 (version 3.3.3) [49].

For the small RNA sequencing data, quality control by FastQC [41] and MultiQC [42] was performed before and after adapter and quality trimming by Cutadapt (version 1.18) [43]. QuickMIRSeq pipeline was used for miRNA sequencing analysis [50]. Reads with lengths of less than 16 and more than 28 were removed. Reads were mapped to the miRBase version 21 to obtain the read counts and RPM values for each miRNA. An optional “Remapping” step was used to map miRNA sequences with mismatches to the reference human genome to reduce the number of likely false positives. Potentially noisy reads were filtered out by removing reads with an average number per sample of 2 or less and missing in at least 90% of samples. IsomiR analysis was performed based on the QuickMIRSeq data. Differential expression analysis of miRNAs was conducted using EdgeR (version 3.36.0) [51]. The method of the trimmed mean of M-values (TMM) was applied for the normalization of the library sizes. TMM-normalized counts per million (CPM) data from edgeR were used through filtering by maximal mean in any group > 10 CPM. Differentially expressed miRNAs among the study groups were identified using the quasi-likelihood F-test with FDR-adjusted *p*-values < 0.05 and absolute log_2_FC values > 1. In all comparisons, the fold change (FC) refers to the ratio of expression in fast fibers relative to slow fibers (type2/type1). PCA was performed by the PCAtools R package (version 2.6.0) [48] using non-scaled TMM-normalized log_2_CPM values.

Transcription factor analysis was performed with the ChIP-X Enrichment Analysis 3 (ChEA3) online tool [52]. 

To find correlations between the expression of miRNAs and their host or target genes, Spearman correlations and corresponding two-sided *p*-values were calculated using matCorr and matCorSig functions of the DGCA R package (version 1.0.2) [53]. The targets for miRNAs were found according to the miRTarBase [54], considering only interactions that were experimentally validated by at least three different methods: reporter assay, Western blot and qPCR. The permutation *p*-values for the correlations were obtained by means of 1000 permutations between samples and calculating the fraction of the absolute values of correlations, which were at least as extreme as those obtained from the original data.

## 3. Results

### 3.1. Muscle Fiber Composition in the Study Groups

The gene expression study cohort collected for the whole transcriptome analysis (*N* = 24, mean age ± SD: 32.7 ± 8.9 years) included two groups of equal size with a predominance of slow-twitch or fast-twitch fibers in the *vastus lateralis* skeletal muscle samples. The first group (type1, *n* = 12; mean age ± SD: 35.0 ± 10.1 years) included physically active individuals or endurance athletes with a predominance of slow-twitch muscle fibers (60.8–94.1%, mean ± SD: 72.5 ± 9.5%). The second group (type2, *n* = 12; mean age ± SD: 30.9 ± 7.4 years) included physically active individuals or power athletes with a predominance of fast-twitch muscle fibers (64.5–80.7%, mean ± SD: 69.8 ± 4.5%). The metadata for these groups are provided in Table A1, Appendix B. 

For the miRNA study, ten male athletes under the age of 40 years were selected from the gene expression study cohort. Five endurance athletes with a predominance of slow-twitch muscle fibers of more than 60% (61.6–72.8%) were selected from the type1 group. Five power athletes with a predominance of fast-twitch muscle fibers of more than 69% (69.3–80.7%) were selected from the type2 group. The age, height, weight and training background of the participants are presented in Table 1. As expected, the proportion of slow-twitch muscle fibers was significantly (*p* < 0.0001) higher in endurance athletes (68.4 ± 5.0%, range 61.6–72.8%) compared to power athletes (30.3 ± 5.3%, range 22.1–35.4%). On the other hand, power athletes (72.3 ± 4.9%, range 69.3–80.7%) had significantly (*p* < 0.0001) more fast-twitch muscle fibers than endurance athletes (34.9 ± 4.4%, range 30.0–39.6%).

### 3.2. Transcriptomic Differences between Groups with a Predominance of Fast- and Slow-Twitch Muscle Fibers

The whole transcriptome analysis was performed for 24 samples of the gene expression study cohort (GEO accession number GSE200398). Our sample prefixes “s” correspond to prefixes “VL017_” in the GEO dataset version.

One sample (s024) was excluded from the analysis due to a low number of reads compared to the other samples. The number of reads per sample in the transcriptome analysis ranged from 40.8 M to 59.3M (mean 48.4 M, SD 4.8 M, *N* = 23).

After filtering and quality control (see the Materials and Methods section for details), the total number of protein-coding genes expressed was 9086. PCA based on the expression of protein-coding genes from the RNA-seq data showed clear separation of clusters including samples with a predominance of type I and type II fibers on the PC1-PC2 plot (Figure 1). With the criteria of |log_2_FC| > 0.585 (changes more than 1.5-fold) and FDR < 0.05, we identified 352 differentially expressed genes, among which 180 were upregulated and 172 were downregulated in the predominance of fast muscle fibers (all genes with FDR < 0.05 are listed in Appendix A). Based on studies of transcriptomic and proteomic differences between type I and type II muscle fibers [7,9], we selected 35 protein-coding genes belonging to three groups based on their expression being (1) specific to a particular fiber type, (2) having differences between slow and fast fibers, and being (3) specific for skeletal muscles but without the expected differences between fiber types (Table A2, Appendix B). Genes were classified into three main groups according to their biological origin and role: (1) myofibrillar structural proteins, (2) proteins related to excitability and excitation–contraction (E–C) coupling and (3) proteins related to general enzymatic activity. Mitochondrial protein genes were not included in this list; however, their expression was expectedly elevated (from 1.4- to 1.8-fold, with average 1.6-fold for 12 genes with FDR < 0.05) in slow fibers because of the higher number of mitochondria and increased oxidative activity. Based on this gene list, we built a heatmap of their expression in the study groups (Figure 2). Based on the expression of these 35 genes, the samples were grouped according to the predominance of a particular fiber type, except for a few individual samples. Almost all genes from groups specific to a particular fiber type or prevalent in slow or fast fibers were differentially expressed among the study groups (Figure 3). Genes characteristically expressed in the slow fibers (*MYH7*, *MYH7B*, *MYL2*, *MYL3*, *TNNC1*, *TNNI1*, *TNNT1*, *TPM3*, *ATP2A2*, *CASQ2* and *LDHB*) were upregulated in the slow-fiber-dominated group. Genes characteristically expressed in the fast fibers (*MYH1, MYH2, MYLPF*, *MYBPC2*, *TNNC2*, *TNNI2*, *TNNT3*, *TPM1*, *ATP2A1* and *LDHA*) were upregulated in the fast-fiber-dominated group. Interestingly, two genes for which no differences in expression between muscle fiber types were expected (*ACTC1* and *MYBPH*) were strongly upregulated in the group with a predominance of fast fibers. However, the expression of these genes was very low, and the differences between the groups could be a result of random fluctuations among the samples. Overall, the expression of characteristic proteins associated with muscle fiber type allowed us to clearly separate the two groups with fast and slow fiber predominance based on the transcriptomic data.

To identify the transcription factors (TFs) responsible for the observed changes in gene expression, we submitted the sets of DE up- and down-regulated genes (180 and 172 genes for type2/type1 ratio, respectively) to the online ChEA3 tool and obtained top 10 TFs based on their mean rank (average integrated rank across all TF-target gene set libraries) for each group of genes (Table 2). The lists of TFs for up- and down-regulated genes did not overlap. We observed a high similarity in TF expression levels in skeletal muscle between our data and those from the Genotype-Tissue Expression project (GTEx) (Table 2). Some of the top 10 TFs were not expressed in skeletal muscle either according to our data or according to the GTEx.

TF analysis showed that the expression of DE genes characteristic of fast fibers is regulated primarily by MRFs: MYOG, MYF6, MYF5 and MYOD1. Of these, MYF5 was not expressed in skeletal muscle, and MYF6 expression was upregulated 1.5-fold in fast fibers. MRFs govern the key processes during myogenesis and are fundamental for skeletal muscle development [55,56]. Additionally, a number of genes characteristic of fast fibers are possibly regulated by YBX3, the only Y-box (YBX) protein detected in human skeletal muscle. Like other members of the YBX family, YBX3 plays diverse roles in biology, including during development, in spermatogenesis and cellular differentiation and proliferation [57,58,59,60,61]. YBX3 emerges as a regulator of large neutral amino acid homeostasis by stabilizing mRNAs of the solute carrier (SLC) amino acid transporters SLC7A5 and SLC3A2 [62]. TF analysis of the list of genes upregulated in fast fibers revealed a relationship with some other TFs, which play an important role in skeletal muscle differentiation, regeneration and growth: MEF2C, PITX2 and TEAD4 [63,64,65,66]. Recently, it has been discovered that myocyte enhancer factor 2C (MEF2C) alters the expression of muscle-specific miRNAs during skeletal muscle differentiation [67].

Among the TFs linked with the list of DE genes characteristic of slow fibers, CHCHD3 had the highest expression level in skeletal muscle tissue. This inner mitochondrial membrane protein is essential for maintaining crista integrity and mitochondrial function [68,69]. The presence of CHCHD3 among the TF enrichment results is probably due to increased expression of mitochondrial proteins—not only those encoded by mitochondrial DNA but also those that play a key role in mitochondrial function, namely NDUFA9, PPTC7, ATP5PB, HADHB, CKMT2, MLYCD, SDHB, LDHB, COX7B and COX7A2—in the slow fiber type. Among the other top 10 TFs with non-zero expression, RORC is induced during skeletal muscle cell differentiation, while DMRT2 and DPF3 are involved in early muscle development and somite patterning [70,71,72].

### 3.3. Presence and Diversity of miRNAs in Small RNA Sequencing Data

Raw small RNA sequencing data for ten samples were deposited to NCBI’s Sequencing Read Archive (SRA) with BioProject accession number PRJNA887354. The general data of miRNA sequencing analysis are presented in Figure 4 and are additionally provided in Table A3, Appendix B. The average number of reads per sample considering the settings and filters in QuickMIRSeq tool was 1.09 M (0.91 to 1.53 M), with the mean ratio of miRNA reads being 95% (92.8 to 96.4%). The total number of miRNAs detected within the sample varied from 345 to 395.

The most abundant miRNA was muscle-specific miR-1-3p, which accounted for an average of 79% of all miRNA reads (77 to 81%, Figure 5A). Figure 5B shows the top 15 expressed miRNAs excluding miR-1-3p across all samples based on CPM values.

Further analysis included 181 miRNAs with a mean TMM-normalized CPM value ≥10 in at least one of the study groups.

Since mature miRNAs are usually presented in various isoforms (isomiRs) resulting from their preprocessing, we analyzed the sequences of miRNA reads to identify the diversity of isoforms in the skeletal muscle tissue samples according to the recent isomiR classification [73]. Isoforms with very low prevalence (mean read count of isoform < 10) were filtered out. After filtering, 1761 isoforms remained for 124 miRNA species. The percentage of reads that matched exactly the canonical mature miRNA sequences was similar to the percentage of isoforms that had modifications on the 3′-end (43 and 45%, respectively, Figure 6A). Among 3′-modifications, the proportion of 3′-trimming and 3′-extension was similar. 5′-end modifications were present in ~5% of the sequences. Polymorphic modifications (i.e., those with single-nucleotide polymorphisms compared to the canonical miRNA sequence) were present in ~15% of the sequences. The percentage distribution of isomiRs was highly similar in the study samples (Figure 6B) and did not differ between the study groups. The sequences and counts of the most commonly represented isoforms accounting for more than 10% of the expression of the corresponding miRNA are listed in Appendix A (207 isoforms for 124 miRNAs, including canonical sequences).

### 3.4. Differential Expression of miRNAs between Groups with Predominance of Fast- and Slow-Twitch Muscle Fibers

PCA showed that the groups with a predominance of fast- and slow-twitch muscle fibers were separated by miRNA profiles in the PC1-PC2 plot (Figure 7A). The main contributors to both PC1 and PC2 were miR-499a-5p, miR-208b-5p, miR-206 and miR-183-5p (Figure 7B).

We found five differentially expressed (DE) miRNAs (Table 3 and Figure 8). Upregulation of miR-206, miR-501-3p and miR-185-5p, and downregulation of miR-499a-5p and miR-208-5p was found in the group with fast-twitch fiber prevalence compared to the group with slow-twitch fiber prevalence (Figure 9 and Figure 10).

The first three miRNAs in Table 3 belong to the group of muscle-specific miRNAs known as myomiRs. Among the detected DE miRNAs, only miR-206 belonged to the group with canonical biogenesis. Another four of the DE miRNAs originated from miRtrons, i.e., their expression is linked with the expression of their host genes in the genome (see Table 4). The precursors of these miRNAs originate from the introns of these genes, representing the non-canonical pathway of miRNA biogenesis [74].

### 3.5. Interactions between MiRNome and Transcriptome

We performed Spearman’s correlation analysis to find the interactions between the transcriptomic and miRNomic data and to link the gene expression patterns for slow-twitch and fast-twitch muscle fibers with the miRNA expression. Since most of the differentially expressed miRNAs originated from miRtrons, we calculated the correlation between the expression of these miRNAs and their host genes based on the total RNA-seq data (Table 4). Two miRNAs (miR-208b-3p and miR-499a-5p) had strong positive correlations with the expression of their host genes (*MYH7* and *MYH7B*, respectively). The expression of both of these genes is characteristic for slow-twitch muscle fibers and was elevated two-fold in samples with a predominance of slow fibers.

We also calculated the correlation between the expression of miRNAs and their potential target genes. Strong positive and negative correlations with the absolute value of correlation coefficient > 0.8 are shown in Table 5. We found 11 strong negative correlations, 2 of which belonged to differentially expressed miRNAs, miR-208b-3p and miR-499a-5p, and indicated their possible regulation of the expression of genes *CDKN1A* and *FOXO4*, respectively. Using the MIENTURNET web tool [75], we built the interaction network between miRNAs and their experimentally validated mRNA targets with at least two shared miRNA–target interactions and the adjusted *p*-value (FDR) ≤ 0.1 (Figure 11). The network shows that the two miRtronic miRNAs specific to slow fibers, miR-208b-3p and miR-499a-5p, share the highest number of common targets and probably have similar regulatory functions, as their seed region sequences are identical in the first six nucleotides.

## 4. Discussion

This study was the first to describe the differences in miRNA profiles in human skeletal muscle with distinct fiber type composition using high-throughput small RNA sequencing. For this study, a unique set of athlete *m. vastus lateralis* tissue biopsy samples were collected in which the proportion of fast- and slow-twitch fibers had marginal values, which deviated from the average expected ratio for that muscle type [76,77].

The number and quality of small RNA reads in the sequencing were sufficient to confidently analyze the expression of 181 miRNAs after the low-expression species cutoff. In all samples, miRNAs represented the major fraction of small RNAs. The representation of miRNA isoforms corresponded to the biologically expected distribution [73] and did not differ among the study groups.

To assess the relevance of our results on miRNA composition in human skeletal muscle, we compared our data with previously obtained miRNA sequencing datasets, which included samples from male *vastus lateralis* muscle. We found four datasets with the ability to download raw sequencing data from the Sequencing Read Archive (SRA) (Table 6). In all datasets, sequencing was performed in a single-end mode using the Illumina platform. Comparison sample sets included only samples from men under 50 years collected before any exercise training. The data were analyzed in the same way as for this study. Each miRNA was ranked according to its representation in muscle tissue based on the mean CPM value in each of the datasets examined. A heatmap visualization of this comparison is presented in Figure 12 for the miRNAs representing the top 25 in this study. In three of the five datasets analyzed, including our study, miR-1-3p and miR-133a-3p were the first and second most represented, with miR-133a-3p being in the top three in all datasets analyzed. The comparison shows that one of the studies (dataset 1) had a similar miRNA profile to our study. Both of these studies used the same protocol to prepare small RNA libraries for sequencing (Table 6). Skeletal muscle miRNA profiles in two other studies (datasets 2 and 3, which used another protocol for small RNA library preparation) were similar but differed from our study. The fourth study (dataset 4) used a custom protocol to prepare miRNA libraries and had a different miRNA profile from all other studies. It can be assumed that the choice of library preparation technique for sequencing influenced the final miRNA ratio. Small RNA library preparation methods may introduce serious bias, mainly during adapter ligation steps followed by reverse transcription and PCR amplification. Read numbers may not reflect actual miRNA expression levels, and different miRNAs may be either over- or under-represented in the library [78]. In addition, miRNA profiles may differ from study to study due to a combination of pre-analytical and biological reasons, such as the methodology of small RNA isolation, the ratio of different cell populations and muscle fibers, as well as the amount of adipose tissue and blood in the biopsy samples.

By studying the transcriptomic data on an extended set of muscle tissue samples, which included groups with a predominance of fast and slow fibers, we found that most genes characteristic of a particular fiber type have statistically significant differences in expression by more than 1.5-fold. The expression of these genes accounts for the differences in both mechanical and energetic properties of different muscle fiber types. The study groups were clearly distinguishable by the transcriptomic data, even though the samples were not pure type I and type II fibers but always a mixture of these types in different ratios. The results of gene expression analysis are in high concordance with previous transcriptomic and proteomic studies, which revealed gene and protein expression patterns characteristic for each fiber type [7,8,9]. TF analysis showed that the gene expression patterns characteristic of fast and slow fibers are mainly regulated by the distinct non-overlapping sets of TFs associated with skeletal muscle development and differentiation, with MRFs being primarily regulators for genes overexpressed in fast fibers. TF analysis supports the assumption that during muscle development, the primary myoblasts have a slow phenotype, and a fast phenotype is developed in secondary myoblasts under the influence of MRFs [83]. However, this is only a simplification and does not reflect the whole picture of muscle fiber type formation and maintenance, which involves different metabolic pathways and is orchestrated by various regulatory factors, including miRNAs [3,36,83,84,85,86,87].

The main finding of this study is the difference in the miRNA expression between muscle tissue samples with different fiber type composition. We found that the expression of miR-208b-3p and miR-499a-5p was upregulated in samples with a predominance of slow-twitch fibers. This is consistent with some previous studies performed on vertebrate species [33,34,35,36,37,38]. The upregulation of these muscle-specific miRNAs in slow-twitch fibers is most likely due to the increased expression of their host genes *MYH7* and *MYH7B*, which was described earlier [33] and was confirmed by the correlation analysis in this study. MiR-208b and miR-499 belong to the miR-208 family, have similar seed-region sequences, are functionally redundant and play a dominant role in the specification of muscle fiber identity [33,88]. Their actions are mediated by a collection of transcriptional repressors of slow myofiber genes, such as Sox6 [33]. It has been demonstrated that type I muscle fiber proportion is increased via the stimulatory actions of estrogen-related receptor γ (ERRγ) on the expression of miR-499 and miR-208b [89]. These miRNAs share 36 common experimentally validated target genes according to miRTarBase (network in Figure 11). We found that the expression of miR-208b-5p and miR-499-5p was negatively correlated with the expression of their target genes *CDKN1A* and *FOXO4*, respectively. It has been shown that miR-208b could regulate cell cycle and promote cattle primary myoblast cell proliferation by targeting CDKN1A [90]. Previously, we found that the *CDKN1A* gene expression was positively correlated with the percentage of fast-twitch muscle fibers in the human *vastus lateralis* muscle [39]. In our study, *CDKN1A* expression was upregulated in samples with a predominance of fast-twitch fibers (log_2_FC = 2.0, FDR < 0.05). *CDKN1A* encodes a cyclin-dependent kinase inhibitor 1A (also known as p21), which is involved in cell cycle regulation (including stem cell proliferation), transcription, apoptosis, DNA repair and cell motility [91]. Notably, *CDKN1A* locus is associated with muscle fiber composition according to a recent genome-wide association study [39]. Moreover, subsequent studies confirmed that miR-208b could mediate skeletal muscle development and energy homoeostasis through specific targeting of TCF12 and FNIP1 [92] and could regulate the conversion of skeletal muscle fiber types by inhibiting METTL8 expression [93]. All these observations indicate that increased expression of miRNAs from the miR-208 family is expected in the predominance of slow fibers. Although the presence of these miRNAs depends on the expression of their host genes, their regulatory functions include activation of slow and repression of fast myofiber gene programs.

We found that the expression of miR-206, miR-501-3p and miR-185-5p was upregulated in samples with a predominance of fast-twitch fibers. MiR-206 is the skeletal muscle-specific myomiR, which had the highest expression level among the DE miRNAs in this study. It performs a variety of regulatory functions in the myoblast proliferation and differentiation by participating in different regulation pathways, in particular by repressing PAX7 or by targeting the *Notch3* gene [28,94,95]. Although miR-206 shares close sequence similarity to miR-1, it has been experimentally proven that miR-206 alone is important for differentiation of myoblasts to myotubes [96]. A recent study on mice with triple knockout of miR-206, miR-1a-1 and miR-1a-2 showed that the miR-206 family is not absolutely essential for myogenesis and is instead a modulator of optimal differentiation of skeletal myoblasts [97]. Studies using small RNA sequencing to assess miRNA expression in muscle fibers of different vertebrate species have shown upregulation of miR-206 in both fast and slow fibers [35,37]. The most recent study in mice showed that miR-206 enforces a slow muscle phenotype [98]. In our study, the expression of miR-206 was upregulated about three-fold in samples with a predominance of fast fibers and had negative correlation with the *NOTCH3* expression (Spearman’s correlation coefficient −0.782). Thus, there may be differences in miR-206 expression patterns in muscle fibers between humans and other vertebrate species, which could be the subject of further investigation.

MiR-501-3p and miR-185-5p have not been previously mentioned in relation to muscle fiber type composition. MiR-501 is located in an intron of isoform-2 of the *CLCN5* gene and is expressed specifically in activated myogenic progenitors and newly formed myofibers [99]. MiR-501-3p has been discovered as a novel muscle-specific miRNA regulating myosin heavy chain during muscle regeneration, and it forms a feedback loop with FOS, MDFI and MYOD to regulate C2C12 myogenesis [99,100]. MiR-185-5p is also an intronic miRNA located within the *TANGO2* gene. We found no differential expression of this gene among the study groups, nor did we find a correlation of its expression with miR-185-5p. Recent data have shown that miR-185-5p targets the apelin receptor, induces collagen production and promotes myocardial fibrosis [101]. Therefore, our new data on the relationship between these two miRNAs and the muscle fiber composition have some experimental evidence supporting their potential role in the regulation of muscle function.

The main limitation of the study is the small sample size for miRNA analysis. This is due to the restrictive criteria we used to form a set of samples from the gene expression cohort; only samples from power and endurance athletes up to age 40 with marginal muscle fiber ratios were included. The transcriptome–miRNome pair database from this study will be the second obtained for human muscle biopsy samples along with the FUSION Tissue Biopsy Study (BioProject PRJNA306562) where the percentage of muscle fibers is characterized. However, despite the small sample size, we showed statistically significant differences in miRNA expression between the study groups and substantiated their biological relevance.

Another limitation of the study is that the biological samples obtained are not pure muscle fibers of types I and II; instead, they are always a mixture of them in a certain ratio. In addition, some deviations in the miRNA profiles may be due to contamination of other cell types, adipose tissue or blood. The methodology of this study did not involve additional analysis of the influence of these factors. However, the results of the transcriptome analysis allowed us to clearly separate the groups by gene expression characteristic of each of the two fiber types.

In conclusion, in this study, we used small RNA sequencing to identify five miRNAs, which are associated with the proportion of fast and slow fibers in the human *vastus lateralis* skeletal muscle. Using transcriptomic data, we showed that the groups of samples with a predominance of fast and slow fibers are clearly differentiated by gene expression. Based on the combined analysis of the miRNome and transcriptome, we can conclude that the differences in miRNA expression are explained to a greater extent by the expression of related genes than vice versa. This is primarily due to the non-canonical miRtronic pathway of miRNA biogenesis and their relationship with the expression of their host genes. However, these miRNAs affect the expression of several genes involved in myogenesis and muscle differentiation, which was also confirmed in our study based on miRNA–mRNA expression correlations. For each of the differentially expressed miRNAs found, a role in the muscle development and function was confirmed by previous studies, which seems to allow us to distinguish them as true biomarkers of skeletal muscle fiber composition.

## Figures and Tables

**Figure 1 life-13-00659-f001:**
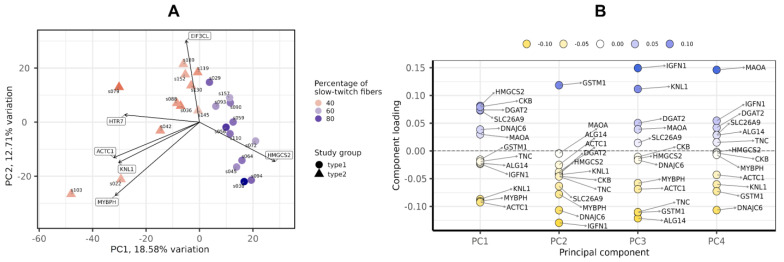
Principal component analysis of transcriptomic profiles by the PCAtools R package, version 2.2.0. (**A**) PC1-PC2 plot showing the study groups and the percentage of slow-twitch muscle fibers. (**B**) Component loadings for the first four PCs. PC, principal component.

**Figure 2 life-13-00659-f002:**
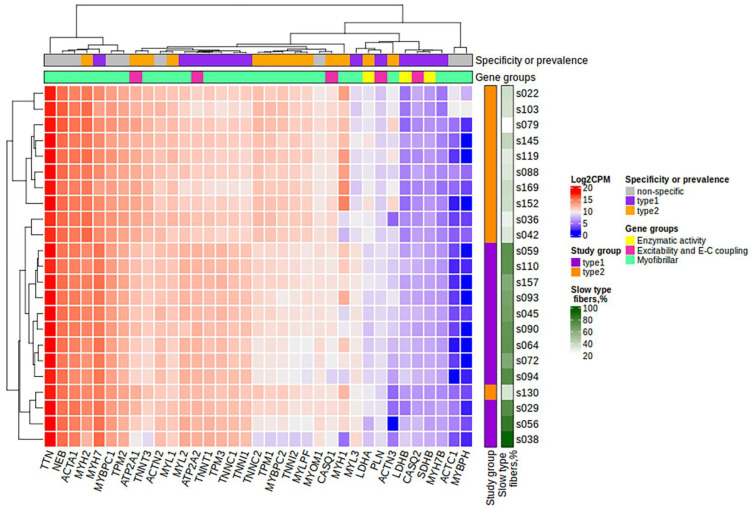
The heatmap of the expression of 35 protein-coding genes belonging to three groups: (1) those specific to a particular fiber type, (2) those with differences between slow and fast fibers, and (3) those specific or characteristic for skeletal muscles but without the expected differences between fiber types. Specificity or prevalence of the gene expression is marked as “type1” for genes expressed in slow-twitch (type I) fibers, “type2” for genes expressed in fast-twitch (type II) fibers and “non-specific” for genes specific or characteristic for skeletal muscles but without the expected differences between fiber types. The heatmap was built using the log_2_CPM values. CPM, counts per million; E-C, excitation–contraction.

**Figure 3 life-13-00659-f003:**
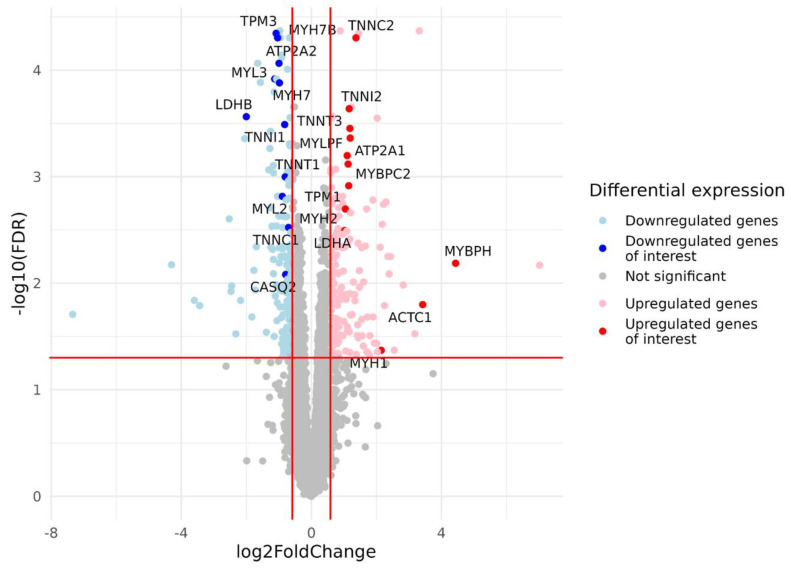
Volcano plot based on statistical comparison of the gene expression (quasi-likelihood F-test and 5% FDR correction using EdgeR) between the study groups (*N* = 23). Each dot represents one protein-coding gene. Up- and down-regulated genes are genes with increased and decreased expression in the group with a predominance of fast-twitch fibers (type2) compared to the group with a predominance of slow-twitch fibers (type1). Log_2_FC value represents type2/type1 CPM ratio. Differentially expressed genes with both FDR < 0.05 and |log_2_FC| > 0.585 are marked in blue (downregulated genes) or red (upregulated genes). For the genes of interest (Table A2, Appendix B), dots are highlighted, and gene names are provided. Red lines indicate FDR and log_2_FC thresholds. FDR, false discovery rate; FC, fold change.

**Figure 4 life-13-00659-f004:**
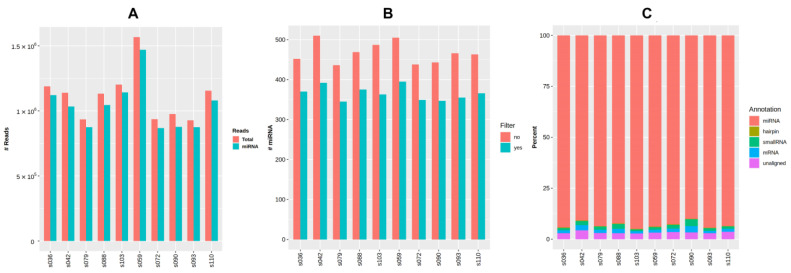
General output of miRNA sequencing analysis using the QuickMIRSeq pipeline. (**A**) Total and miRNA reads in each sample. (**B**) Distribution of small RNA reads by type. (**C**) Number of detected miRNAs per sample with and without filtering of noisy reads.

**Figure 5 life-13-00659-f005:**
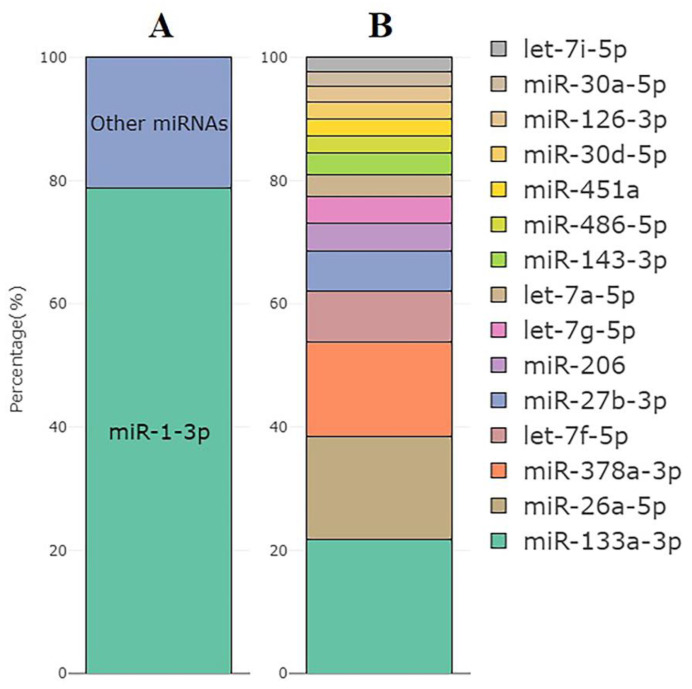
(**A**) The mean percentage of miR-1-3p reads within all miRNA reads in the set of samples used for the miRNA analysis (*n* = 10). (**B**) Top 15 miRNAs with the highest expression excluding miR-1-3p and their distribution by the CPM values.

**Figure 6 life-13-00659-f006:**
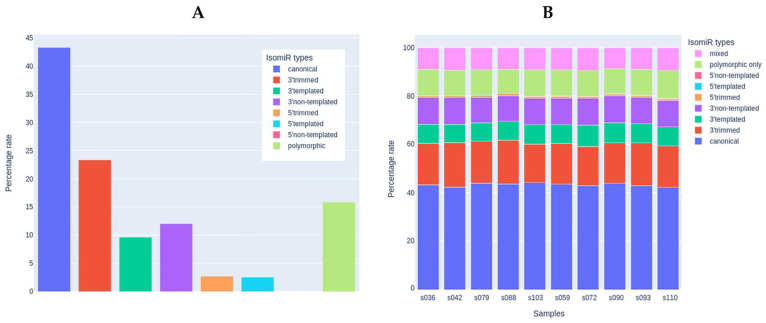
The analysis of miRNA isoforms (isomiRs). (**A**) Presence of different isomiR types in the set of samples used for the miRNA analysis (*n* = 10). (**B**) Percentage of each isomiR type in each study sample.

**Figure 7 life-13-00659-f007:**
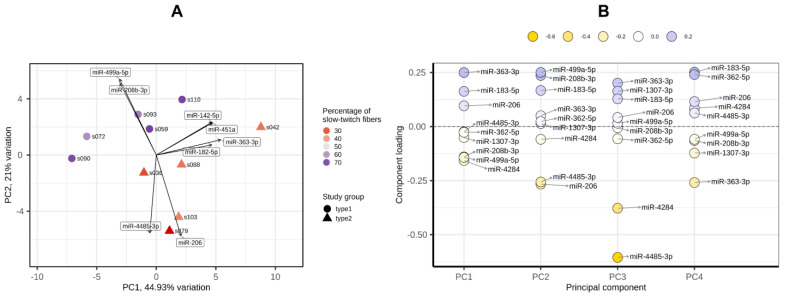
Principal component analysis of miRNA profiles by the PCAtools R package, version 2.2.0. (**A**) PC1-PC2 plot showing the study groups and the percentage of slow-twitch muscle fibers. (**B**) Component loadings for the first four PCs. PC, principal component.

**Figure 8 life-13-00659-f008:**
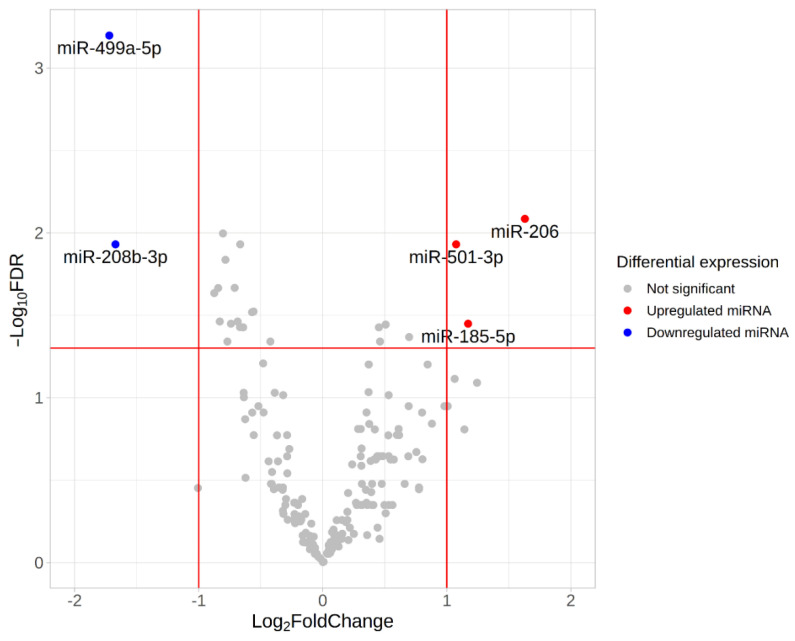
Volcano plot based on statistical comparison of the miRNA expression (quasi-likelihood F-test and 5% FDR correction using EdgeR) among the study groups *(n* = 10). Each dot represents one miRNA. Up- and down-regulated genes are genes with increased and decreased expression in the group with a predominance of fast-twitch fibers (type2) compared to the group with a predominance of slow-twitch fibers (type1). Log_2_FC value represents type2/type1 CPM ratio. Differentially expressed miRNAs with both FDR < 0.05 and |log2FC| > 1 are marked in blue (downregulated miRNAs) or red (upregulated miRNAs). Red lines indicate FDR and log_2_FC thresholds. FDR, false discovery rate; FC, fold change.

**Figure 9 life-13-00659-f009:**
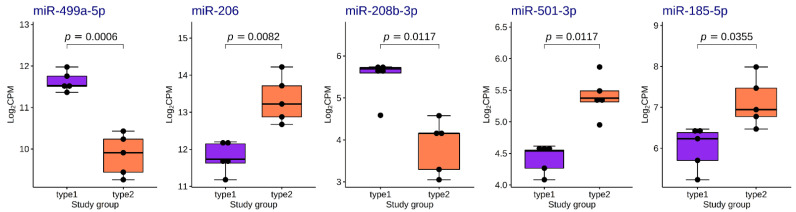
Boxplots of distribution of TMM-normalized log_2_CPM values for differentially expressed miRNAs in the study groups. The boxplots represent median and interquartile ranges (IQRs) in the box and values for individual samples in the dots. Adjusted *p*-values of statistical significance are provided for the pairwise quasi-likelihood F-test comparisons performed in EdgeR.

**Figure 10 life-13-00659-f010:**
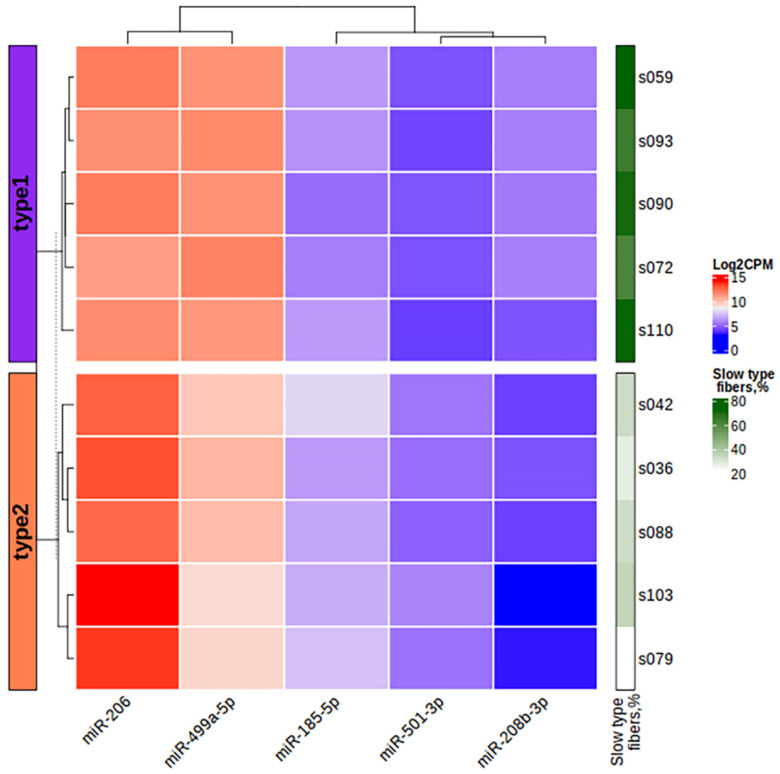
The heatmap of miRNA expression based on the log_2_CPM values of differentially expressed miRNAs in the study samples. The left bar chart indicates the study groups. CPM, counts per million.

**Figure 11 life-13-00659-f011:**
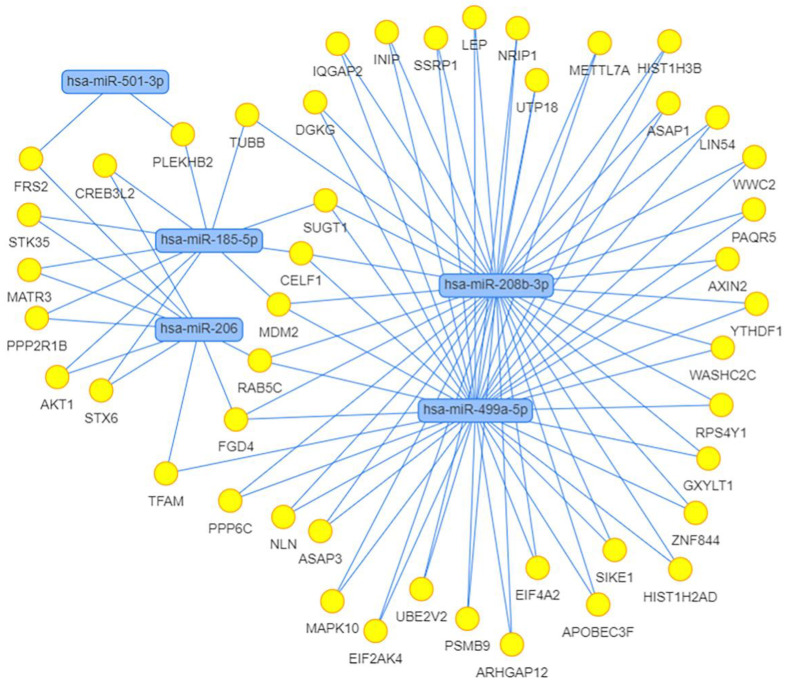
Network of differentially expressed miRNAs and their target genes obtained using MIENTURNET web tool. Targets were selected in the miRTarBase with at least two shared miRNA–target interactions and the adjusted *p*-value (FDR) ≤ 0.1.

**Figure 12 life-13-00659-f012:**
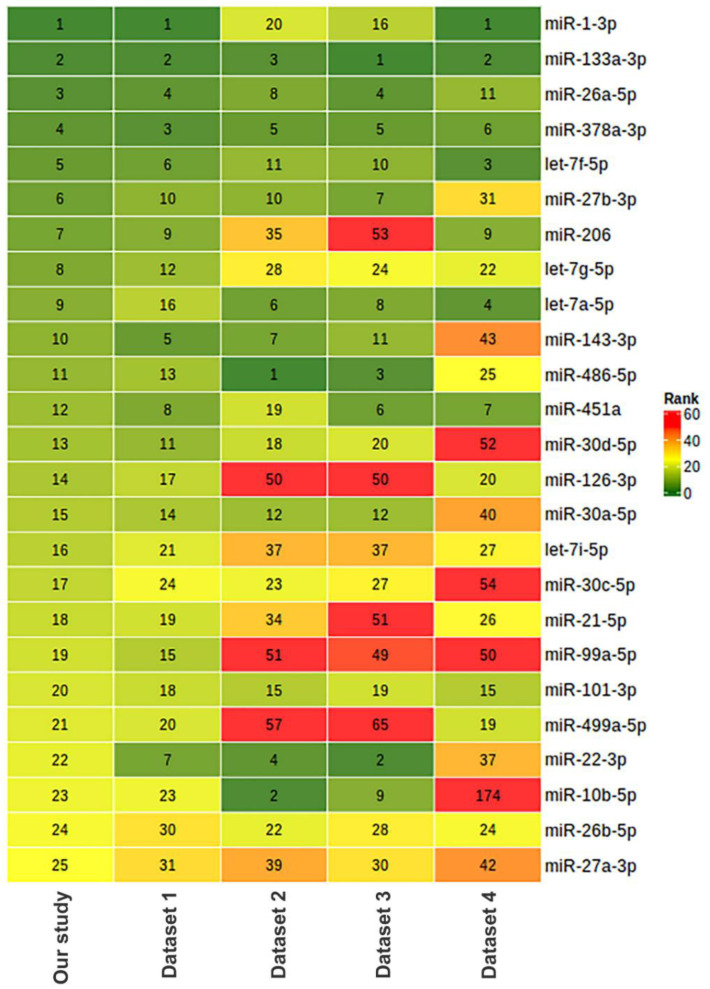
Comparison of miRNA composition in human *vastus lateralis* skeletal muscle in our study and five different studies based on miRNA sequencing. The relative abundance of miRNAs is represented as ranks assigned on the basis of counts per million (CPM) values. The characteristics of the datasets and number of samples taken for the comparison are provided in Table 6. Dataset 1: PRJNA276561 [79], *n* = 5; dataset 2: PRJNA403822 [80], *n* = 10; dataset 3: PRJNA306562 (FUSION Tissue Biopsy Study dataset) [81], *n* = 10; dataset 4 [82]: PRJNA524317, *n* = 3.

**Table 1 life-13-00659-t001:** Characteristics of the participants included in the miRNA study.

ID	Age	BMI	Sports Category	Sports Type	Sports Experience, Years	Percentage of Muscle Fibers, %		Group
						Slow	Fast	
s059	24	19.3	endurance	Mountain running	12	72.8	30.8	type1
s072	26	22.3	endurance	Marathon running	20	61.6	39.6	type1
s090	24	19.7	endurance	Long-distance running	10	70.9	35.4	type1
s093	25	22.1	endurance	Long-distance running	8	64.4	38.9	type1
s110	29	23.2	endurance	Triathlon	12	72.1	30.0	type1
s036	29	26.3	power	Weightlifting	5	27.9	72.4	type2
s042	27	29.9	power	Powerlifting	10	33.1	69.6	type2
s079	39	34	power	Powerlifting	4	22.1	80.7	type2
s088	27	22.2	power	Decathlon	16	32.9	69.3	type2
s103	25	30.1	power	Bodybuilding	8	35.4	69.6	type2

**Table 2 life-13-00659-t002:** Transcriptional factor (TF) analysis for the up- and down-regulated genes: results from ChIP-X Enrichment Analysis 3 (ChEA3). Top 10 TFs by their mean rank are shown for the sets of up- and down-regulated genes. Mean log_2_CPM is given for the study cohort used for the whole transcriptome analysis (*n* = 23). N/A indicates that gene expression did not pass the initial filter for low-expressed genes. Median TPM is given for the GTEx data for skeletal muscle tissue (*N* = 803). CPM, counts per million; TPM, transcripts per million; GTEx, the Genotype-Tissue Expression project.

Rank	Upregulated Genes (*N* = 180)					Downregulated Genes (*N* = 172)				
	TF	Score by Mean Rank	Number of Overlapping Genes	Mean log_2_CPM	Median TPM in GTEx	TF	Score by Mean Rank	Number of Overlapping Genes	Mean log_2_CPM	Median TPM in GTEx
1	MYOG	2.25	69	4.87	58.0	NKX25	1.5	32	N/A	<0.1
2	MYOD1	5.2	77	4.29	21.7	NKX26	18	26	N/A	<0.1
3	MYF5	10.33	43	N/A	2.1	CHCHD3	29.5	18	7.10	89.8
4	MYF6	10.33	35	6.32	255.0	GATA4	40.8	64	N/A	<0.1
5	YBX3	25.5	22	10.68	2214.0	RORC	41.33	23	5.62	47.7
6	FOSL1	26.6	40	N/A	1.4	IRX6	43.67	21	N/A	1.5
7	MEF2C	38.8	49	9.17	34.6	DMRT2	49	21	1.92	0.9
8	PITX2	43.67	37	4.49	16.0	RXRG	54.33	23	4.13	15.5
9	PRRX2	50	23	N/A	0.8	DPF3	54.33	27	4.56	9.2
10	TEAD4	56	73	5.20	59.4	CEBPA	57.2	56	N/A	6.2

**Table 3 life-13-00659-t003:** Differentially expressed miRNAs (both FDR < 0.05 and |log2FC| > 1 in the group comparison using quasi-likelihood F-test with 5% FDR correction performed in EdgeR). Log_2_FC value represents type2/type1 CPM ratio. FC, fold change; CPM, counts per million; FDR, false discovery rate.

miRNA	log_2_FC	log_2_CPM	*p*-Value	FDR-Adjusted *p*-Value
miR-499a-5p	−1.72	11.03	0.000004	0.000634
miR-206	1.63	12.86	0.000091	0.008210
miR-208b-3p	−1.67	4.94	0.000281	0.011725
miR-501-3p	1.08	5.02	0.000334	0.011725
miR-185-5p	1.17	6.77	0.002992	0.035545

**Table 4 life-13-00659-t004:** Correlations in expression between differentially expressed miRtronic miRNAs and their host genes. For the host gene expression, log_2_FC values represent type2/type1 CPM ratio, and *p*-values are provided for differential expression analysis. CPM, counts per million; FC, fold change; FDR, false discovery rate.

miRNA				Host Gene					Spearman’s Correlation Coefficient	Two-Sided *p*-Value
Name	Gene	Chromosome	GRCh38.p13 Coordinates	Name	log_2_CPM	log_2_FC	*p*-Value	FDR-Adjusted *p*-Value		
miR-208b-3p	*MIR208B*	14	23,417,987–23,418,063	*MYH7*	14.63	−0.98	<0.001	<0.001	0.903	<0.001
miR-499a-5p	*MIR499A*	20	34,990,376–34,990,497	*MYH7B*	6.10	−1.04	<0.001	<0.001	0.855	0.002
miR-501-3p	*MIR501*	X	50,009,722–50,009,805	*CLCN5*	2.87	0.54	0.017	0.117	0.564	0.090
miR-185-5p	*MIR185*	22	20,033,139–20,033,220	*TANGO2*	3.21	−0.25	0.336	0.598	0.442	0.200

**Table 5 life-13-00659-t005:** Correlations between the expression of miRNAs and their potential target genes. CPM, counts per million; FC, fold change; FDR, false discovery rate.

miRNA	Target Gene	Spearman’s Correlation Coefficient	Two-Sided *p*-Value	Permutation Test *p*-Value
miR-20a-5p	*FBXO31*	−0.879	0.00081	0.005
miR-152-3p	*MAFB*	−0.867	0.00117	0.005
miR-143-3p	*DNMT3A*	−0.842	0.00222	0.004
miR-17-5p	*FBXO31*	−0.842	0.00222	0.004
miR-208b-3p	*CDKN1A*	−0.842	0.00222	0.005
miR-25-3p	*MTF1*	−0.830	0.00294	0.005
miR-340-5p	*DNMT3A*	−0.830	0.00294	0.005
miR-148a-3p	*RPS6KA5*	−0.818	0.00381	0.001
miR-1-3p	*TAGLN2*	−0.806	0.00486	0.005
miR-186-5p	*HOXA9*	−0.806	0.00486	0.006
miR-499a-5p	*FOXO4*	−0.806	0.00486	0.006
miR-126-3p	*BCL2*	0.806	0.00486	0.010
miR-126-3p	*CRK*	0.806	0.00486	0.009
miR-126-3p	*FOXO3*	0.806	0.00486	0.011
miR-185-5p	*HMGA1*	0.806	0.00486	0.008
miR-21-5p	*TP63*	0.806	0.00486	0.006
miR-22-3p	*ZFP91*	0.806	0.00486	0.006
miR-143-3p	*ITM2B*	0.818	0.00381	0.011
miR-195-5p	*CCND1*	0.818	0.00381	0.009
miR-98-5p	*CASP3*	0.830	0.00294	0.004
miR-126-3p	*KLF10*	0.842	0.00222	0.003
miR-210-3p	*ALDH5A1*	0.842	0.00222	0.004
miR-126-3p	*RHOU*	0.855	0.00164	0.003
miR-199a-5p	*CAV1*	0.867	0.00117	0.005
miR-27a-3p	*ZBTB10*	0.867	0.00117	0.001
miR-210-3p	*HIF3A*	0.879	0.00081	0.002
miR-30b-5p	*CAT*	0.879	0.00081	0.004
miR-499a-5p	*PDCD4*	0.891	0.00054	0.002
miR-30b-5p	*PDGFRB*	0.903	0.00034	0.001

**Table 6 life-13-00659-t006:** Datasets used for miRNA sequencing data comparison. In all datasets, single-end Illumina small RNA sequencing was performed, and small RNA was extracted from *vastus lateralis* muscle biopsy samples from healthy adult participants. Comparison sample sets included samples from male participants under 50 years collected before any exercise training. Manufacturers mentioned: Qiagen, Hilden, Germany; NEB, New England Biolabs, Ipswich, MA, USA; Illumina, San Diego, CA, USA; Thermo Fisher Scientific, Waltham, MA, USA.

Dataset	Authors	NCBI BioProject ID	Small RNA Isolation	Small RNA Library Prep	Illumina Sequencing Platform	Read Length	Muscle Biopsy Samples: Comparison Set/Total Set	Sample Characteristics for the Comparison Set	Reference
1	McLean C.S. et al., 2015	PRJNA276561	miRNeasy kit, Qiagen	NEBNext Small RNA Library Prep Set for Illumina, NEB	HiSeq 2000	50	5/12	Men, biopsies taken before exercise training	[79]
2	Mitchell C.J. et al., 2018	PRJNA403822	AllPrep DNA/RNA/miRNA Universal Kit, Qiagen	TruSeq Small RNA Kit, Illumina	Hiseq 2500	50	10/73	Men, age 40–48, biopsies taken before strength testing	[80]
3	Taylor, D.L. et al., 2019	PRJNA306562	Trizol extraction	TruSeq Small RNA Library Prep Kit v1.5, Illumina	HiSeq 2500	50	10/296	Men, age 25–47	[81]
4	Massart J. et al., 2021	PRJNA524317	mirVana miRNA Isolation Kit, Thermo Fisher Scientific	Custom protocol	Genome Analyzer	36	3/6	Men, biopsies collected before endurance exercise training	[82]
5	Our study	PRJNA887354	miRNeasy kit, Qiagen	NEBNext Small RNA Library Prep Set for Illumina, NEB	MiSeq	50	10/10	Men, age 24–39	-

## Data Availability

Small RNA sequencing data are available in the Sequencing Read Archive repository (BioProject accession number: PRJNA887354). Total RNA sequencing data are deposited in GEO, accession number GSE200398.

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
