# Peer review of "Diversity and Differential Expression of MicroRNAs in the Human Skeletal Muscle with Distinct Fiber Type Composition"

_life, 2023, doi:10.3390/life13030659_

Round 1

Reviewer 1 Report

In this paper Zhelankin et al. investigated the contribution of microRNA  in skeletal muscle which show predominance of slow-twitch or fast-twitch fibers.

The manuscript is well written and the results are interesting.

However, the manuscript is not accepatble in this form since the Reviewer has some important considerations which require major revisions.

The starting point of this paper is the identification of muscle histology, thus the histological analysis of the muscle used for biomolecular studies became very important.

1.    Immunohistochemical staining has been performed on skeletal muscle included in OCT medium to preserve the tissue morphology?

2.    Images of examples of immunostained muscle with predominance of slow-twitch or fast-twitch fibers must be added (low magnification)

3.    As to concern the hybrid fibers, are they excluded or included in the percentage? Could the percentage of hybrid fibers influence the results?

4.    Results: paragraph 3.1, the mean of the percentage of muscle fiber (not only the range) should be added

5.    Results: line 241, the mean age of the second group should be added

Another point is the sample size:

1.    Among the initial 151 muscle samples, 24 were selected for gene expression analysis and 10/24 were selected for miRNA study, a very small sample size. The results show that the two groups with fast and slow fiber predominance are clearly separate based on the transcriptomic data. Thus the Reviewer does not understand why the miRNA was not performed on 24 samples as for gene expression.

Control muscle samples:

1.    Since different methods may introduce serious bias, the comparison with literature data might be difficult. Control muscle samples should be used. Please explain

Due to the limitations of the study, the conclusion sentence might be  “which seems to allow us to distinguish them as true biomarkers of skeletal muscle fiber composition”.

Author Response

We thank the reviewer for the detailed analysis of the manuscript. Below are our responses to the comments.

The starting point of this paper is the identification of muscle histology, thus the histological analysis of the muscle used for biomolecular studies became very important.

  1. Immunohistochemical staining has been performed on skeletal muscle included in OCT medium to preserve the tissue morphology?

No, we did not use the OCT medium. Visual analysis of our serial cross-sections revealed no any defect in skeletal muscle fiber morphology.

  1. Images of examples of immunostained muscle with predominance of slow-twitch or fast-twitch fibers must be added (low magnification)

We have added images of immunostained muscle samples with the predominance of fast-twitch and slow-twitch fibers in low magnification (samples s079 and s110, respectively). Please see Figure A1, Appendix A. We also mentioned this in the text (lines 145-146).

  1. As to concern the hybrid fibers, are they excluded or included in the percentage? Could the percentage of hybrid fibers influence the results?

We used antibodies to slow and fast myosin heavy chain for serial cross-sections. Usually, the sum of relative number of slow and fast fiber (expressed in percent) was slightly greater than 100% (by a few percent) that is related with presence of hybrid fibers. This value was low and had no influence on the results.

  1. Results: paragraph 3.1, the mean of the percentage of muscle fiber (not only the range) should be added

We have added the required information.

  1. Results: line 241, the mean age of the second group should be added

We have added the required information.

Another point is the sample size:

  1. Among the initial 151 muscle samples, 24 were selected for gene expression analysis and 10/24 were selected for miRNA study, a very small sample size. The results show that the two groups with fast and slow fiber predominance are clearly separate based on the transcriptomic data. Thus the Reviewer does not understand why the miRNA was not performed on 24 samples as for gene expression.

The presented work is a preliminary exploratory study, and we initially aimed to use a smaller sample for miRNA analysis compared to a transcriptomic study. We used the restrictive criteria to form a set of samples from the gene expression cohort, including only samples from power and endurance athletes up to age 40 with marginal muscle fiber ratios to form two groups of the same size. Of the 12 samples that met these criteria, two samples (s064 and s169) failed to produce small RNA sequencing libraries, so the final number of samples was 10. Furthermore, we used a single miSeq run with the aim of getting more than 1 million reads per sample, which also limited us in increasing the number of samples. In the manuscript, we describe the small sample size as the main limitation of this study (lines 612-620).

Control muscle samples:

  1. Since different methods may introduce serious bias, the comparison with literature data might be difficult. Control muscle samples should be used. Please explain

Skeletal muscles always contain slow-twitch and fast-twitch fibers, and their ratio varies both between different muscle types and within the same type. Thus, for skeletal muscle, there can be no clear definition of control samples. In our study, we investigate two marginal states of muscle fiber composition, so that each of the compared groups can be a control for the other group.

The comparison of microRNA profiles with other datasets in the Discussion section is not intended to reveal differences in microRNA expression between groups with different muscle fiber composition, as it is not defined in all studies. This comparison is given only to assess whether the skeletal muscle microRNA profile obtained by sequencing in our study is similar or different from that in other similar studies.

Due to the limitations of the study, the conclusion sentence might be “which seems to allow us to distinguish them as true biomarkers of skeletal muscle fiber composition”.

We agree with this comment, so we have changed the conclusion sentence.

Reviewer 2 Report

The manuscript has been done with great effort and hard work observed. Really like the way it is presented!  

Author Response

We thank the reviewer for the appreciation of our manuscript.

Reviewer 3 Report

In this manuscript, the authors profile miRNAs in human participants with distinct fiber types.  They find distinct miRNAs associate with fast type or slow type muscle fibers.  Overall, this is an interesting and informative study and the manuscript is well written.  A limitation of the study is the very small sample size being used, but the authors do demonstrate the statistical significance of their data and address the limitation of the data set very thoroughly in the discussion   Concerns with the manuscript are noted below:

1. The resolution of the figures should be improved for readability (ie, gene names are difficult to read in Figure 1 and 7)

2.  More detail is needed to interpret Figure 3 and Figure 8. The shown genes are downregulated and upregulated with respect to what? Line 290 states that genes expressed in fast twitch fibers were upregulated in the slow twitch fiber dominated group.  Is this sentence correct?

3.  Table 2 shows that MYF5 is third most upregulated gene in the data set and Line 329 states that MYF5 is not expressed in skeletal muscle? The text should be corrected to support the data.

In this manuscript, the authors profile miRNAs in human participants with distinct fiber types.  They find distinct miRNAs associate with fast type or slow type muscle fibers.  Overall, this is an interesting and informative study and the manuscript is well written.  A limitation of the study is the very small sample size being used, but the authors do demonstrate the statistical significance of their data and address the limitation of the data set very thoroughly in the discussion   Concerns with the manuscript are noted below:

1. The resolution of the figures should be improved for readability (ie, gene names are difficult to read in Figure 1 and 7)

2.  More detail is needed to interpret Figure 3 and Figure 8. The shown genes are downregulated and upregulated with respect to what? Line 290 states that genes expressed in fast twitch fibers were upregulated in the slow twitch fiber dominated group.  Is this sentence correct?

3.  Table 2 shows that MYF5 is third most upregulated gene in the data set and Line 329 states that MYF5 is not expressed in skeletal muscle? The text should be corrected to support the data.

Author Response

We thank the reviewer for the detailed analysis of the manuscript. Below are our responses to the comments.

  1. The resolution of the figures should be improved for readability (ie, gene names are difficult to read in Figure 1 and 7)

The resolution of Figures 1 and 7 has been improved (width 10,000 pixels and resolution 320 DPI), the font size for gene names has been increased, and we have minimized the overlap between gene and sample names. We have also increased the font size of the axis labels in Figure 8.

  1. More detail is needed to interpret Figure 3 and Figure 8. The shown genes are downregulated and upregulated with respect to what? Line 290 states that genes expressed in fast twitch fibers were upregulated in the slow twitch fiber dominated group. Is this sentence correct?

We have added descriptions for up- and down-regulated genes to the Figure 3 and Figure 8 captions:

“Up- and down-regulated genes are genes with increased and decreased expression in the group with a predominance of fast-twitch fibers (type2) compared to the group with a predominance of slow-twitch fibers (type1).”

There is indeed an error in line 290, thanks for pointing it out in the text. Changed to “Genes characteristically expressed in the fast fibers (MYH1, MYH2, MYLPF, MYBPC2, TNNC2, TNNI2, TNNT3, TPM1, ATP2A1, and LDHA) were upregulated in the fast-fiber-dominated group”.

  1. Table 2 shows that MYF5 is third most upregulated gene in the data set and Line 329 states that MYF5 is not expressed in skeletal muscle? The text should be corrected to support the data.

Table 2 shows the results of transcription factor analysis. The order of the genes in it does not reflect their expression, but reflects their rank as potential regulatory factors obtained using the ChEA3 online tool.  The expression of these genes is shown in the "Mean log2CPM" column of this table, and “N/A” value means that MYF5 expression is very low and did not pass the initial filter for low-expressed genes. Low expression of MYF5 in human skeletal muscle is also supported by the GTEx data (“Median TPM in GTEx” column of the Table 2). Line 329 in the manuscript is consistent with these observations.

Round 2

Reviewer 1 Report

Thanks to the Authors for the response to the Reviewer comments in line with the requests.

Author Response

(The authors gave the same response as above.)
